# Impact of COVID-19 Infection on Health-Related Quality of Life in the Japanese Population: A Large Health-Insurance-Based Database Study

**DOI:** 10.3390/ijerph21020217

**Published:** 2024-02-13

**Authors:** Tomoko Kobayashi, Chikara Miyaji, Hiroshi Habu, Yoshiharu Horie, Soshi Takao

**Affiliations:** 1Department of Epidemiology, Okayama University Graduate School of Medicine, Dentistry and Pharmaceutical Sciences, 2-5-1 Shikata-cho, Kita-ku, Okayama 700-8558, Japan; pfmx01sj@s.okayama-u.ac.jp (T.K.); miyaji-c@okayama-u.ac.jp (C.M.); pz2g0mcn@s.okayama-u.ac.jp (Y.H.); 2Health Service Center, Okayama University, 2-1-1 Tsushimanaka, Kita-ku, Okayama 700-0082, Japan; 3Department of Social Epidemiology, Graduate School of Medicine and School of Public Health, Kyoto University, Yoshida-konoe-cho, Sakyo-ku, Kyoto 606-8501, Japan

**Keywords:** health-related quality of life, COVID-19 infection, Japan

## Abstract

Evidence for acute or long-term coronavirus disease 2019 (COVID-19) infection is relatively limited. We aimed to evaluate the impact of COVID-19 infection on health-related quality of life (HRQoL) in the Japanese population. Eligible study participants were 13,365 employees and their dependents who answered questionnaires at baseline and 18 months later and who had at least 6 months of continuous enrolment before and after baseline. Of the 711 study participants who developed COVID-19 infection, 29.0% reported a decline in HRQoL, whereas 25.2% of uninfected participants reported a decline. The adjusted odds ratios (95% confidence intervals) for the association between COVID-19 infection and declines in HRQoL in the age categories of less than 30 years, 30s, 40s, 50s, and 60 years or higher were 0.54 (0.15–1.92), 1.70 (1.03–2.81), 1.14 (0.82–1.57), 1.05 (0.77–1.42), and 0.87 (0.46–1.64), respectively. This study demonstrates a differential association between COVID-19 infection and declines in HRQoL by age group. A 1.7-fold increase in the odds of negative changes in HRQoL was observed in only those in their 30s. Further studies are needed to elucidate differences in the impact of COVID-19 infection on HRQoL between younger people such as those in their 30s and the older population.

## 1. Introduction

Coronavirus disease 2019 (COVID-19) is an infectious disease caused by the severe acute respiratory syndrome coronavirus 2 virus [1]. Infected individuals develop both short- and long-term complications [2]. Such complications are usually not limited to clinical symptoms but affect subjective measures such as health-related quality of life (HRQoL). HRQoL reflects how individuals perceive and react to their physical, functional, and mental health status and the nonmedical aspects of their lives such as their job, family, friends, and other life situations [3]. That is, HRQoL is based on individual perception but also related to the social context. Therefore, HRQoL can provide a comprehensive evaluation that encompasses all of the important individual and social aspects of quality of life that are related to health, and the tool has been widely used in health care and clinical research. A comprehensive evaluation using the HRQoL is especially important in the context of the COVID-19 pandemic, which has affected not only individuals but also the social environment (e.g., healthcare access and resources) because of its mandatory restrictions. Indeed, many of the studies published to date as part of a structured review reported a higher impact of COVID-19 infection on HRQoL in acute COVID-19, women, older adults, patients with more severe diseases, and patients from low-income countries [4]. The acute manifestations and complications of COVID-19 are well known in the literature; however, the long-term effects of COVID-19 after recovery or discharge from the hospital have not been well established [5,6,7]. To the best of our knowledge, evidence in the Japanese population for acute or long-term COVID-19 infection is limited. Only one cross-sectional study in 349 patients who recovered from COVID-19 has shown associations between long-term COVID-19 symptoms and reduced HRQoL scores [8]. Moreover, because previous cohort studies in Bangladesh, Italy, and Spain included only patients who had recovered from COVID-19 [7,9,10], the impact of COVID-19 infection on HRQoL was not measured against a non-infected cohort. Other studies have been conducted in the general population and among social or health workers (not limited to patients who were infected with COVID-19) in Germany and Portugal [5,6]. However, because these studies had a cross-sectional design, HRQoL scores at baseline (i.e., pre-pandemic) were not considered. Moreover, none of the studies in Japan assessed individual changes in HRQoL pre- and post-COVID-19 pandemic.

Despite the availability of several HRQoL measurement instruments, the EuroQoL 5-dimensional-5 levels (EQ-5D-5L) is one of the most used questionnaires in clinical and outcomes research. Its use is recommended by the National Institute for Health and Clinical Excellence [11] and Japan’s economic evaluation guideline [12] to evaluate adjusted life-years by weighting patients’ health status (i.e., HRQoL). In the occupational context, employed status is the most common protective factor for both absolute EQ-5D-5L index scores and each EQ-5D-5L dimension [9]. In contrast, unemployed status is strongly associated with an increased risk of all-cause mortality in young people, suicide, and the development of mental illness in the general population [13,14]. Furthermore, given the proven strong association between employment and HRQoL, further assessments of employees are necessary from economic, social, and public health perspectives. However, long-term studies of employees (not limited to health or social workers) who have experienced the consequences of COVID-19 infection are sparse.

In addition, evidence of how the impact of COVID-19 on HRQoL differs with age and which age group experiences the most pronounced impact is inconclusive. Some studies have reported that older age is significantly associated with a decline in HRQoL [7,9,15,16,17], whereas others have observed a U-shaped pattern that suggests that the younger population also experienced a decline in HRQoL following the pandemic [18,19]. Although older generations are considered most vulnerable given their generally declining health status with age, the COVID-19 pandemic may have differentially affected generations. Thus, using secondary data that consisted of health insurance claims, health check-up data, and questionnaire data, we evaluated the association between COVID-19 infection and changes in HRQoL to determine whether the association was modified by age in Japanese workers and their dependents.

## 2. Materials and Methods

Respondents who answered questionnaires at baseline (June 2020) and 18 months after baseline (December 2021) and who had at least 6 months of continuous enrolment in a health insurance association before baseline were included in the study. Of the 13,395 study participants who met the inclusion criteria, those who were infected by COVID-19 before June 2020 were excluded (*n* = 30). Data for eligible study participants (*n* = 13,365) who fulfilled the inclusion and exclusion criteria were analysed. The data were accessed for use in this study on 29 July 2022 and consisted of three secondary data sources: health insurance claims, annual health check-ups, and self-reported information from employees and their non-working dependents who used the kencom application (DeSC Healthcare Inc., Tokyo, Japan). Health insurance claims data records included monthly information about patient demographics, diagnoses in accordance with the International Classification of Diseases and Related Health Problems, 10th Revision (ICD-10), medical procedures, and medications. Health check-up data consisted of annual physical examination results, the measurement of biomarkers, imaging examinations, and questionnaires regarding medical history, comorbidities, the concomitant use of medications, and lifestyle habits (e.g., smoking status, frequency and amount of alcohol consumption, and exercise habits). Self-reported measures such as HRQoL were collected using the kencom application. Data were anonymised under the ‘opt-out’ agreement between the users and health insurance associations. Under the agreement, users are notified of the usage of their data and their right to request deletion of their data.

COVID-19 infection between June 2020 and December 2021 was defined by the ICD-10 diagnostic codes U07.1 and U10 recorded on health insurance claims.

We assessed HRQoL using the Japanese version of the EQ-5D-5L questionnaire [20] given that the tool’s original version is translated into many languages and widely used to assess the multi-dimensional domains of the health and well-being of various populations [21]. The EQ-5D-5L defines health states by examining five dimensions: mobility, self-care, usual activities, pain/discomfort, and anxiety/depression [22]. Each of the dimensions has five levels of response options (no problems, slight problems, moderate problems, severe problems, and unable to/extreme problems). Each response option is given a unique value (score) that is used to calculate a single utility score: the total score of the five dimensions plus a constant term value (0.0609) are subtracted from 1 and ranges from −0.0255 to 0.9391. A negative value indicates a health state worse than death, 0 represents a health state equivalent to death, and 0.9391 represents perfect health [20,21]. The changes in HRQoL were calculated using the scores at baseline and 18 months later. The score was divided into two categories based on whether the change value was zero or positive (not worse than baseline HRQoL) or negative (worse than baseline HRQoL).

Age and sex at baseline and other variables were obtained from annual health check-up data in 2020. The variables that were considered potential confounders of the association between COVID-19 infection and HRQoL based on the existing evidence were included as covariates in the adjusted models: age (continuous) [23,24,25], sex [23,24,25], antihypertensive drug use (yes; no) [25,26,27], insulin injection or hypoglycaemic drug use (yes; no) [25,26,27], cholesterol-lowering drug use (yes; no) [25,26,27], body mass index (BMI) [27], smoking status (yes: smoked more than a total of 100 cigarettes in the past period or for more than 6 months and smoked in the latest month; no) [28], frequency of alcohol consumption (daily, sometimes, rarely) [29], exercise habits (yes: exercise with slight sweating for 30 min or more than at least 2 days/week for 1 year or more; no) [30], and sufficient rest through sleep (yes; no) [31]. BMI was classified into three categories based on the criteria of the Japan Society for the Study of Obesity [32]: underweight (<18.5 kg/m^2^), normal body weight (18.5–25 kg/m^2^), and overweight (≥25 kg/m^2^).

Descriptive statistics were used to summarise the characteristics of the study participants, the proportion of COVID-19 infections, changes in HRQoL, and other variables. We initially tested statistical interactions by using cross-product terms for COVID-19 infection and age categories (less than 30 years, 30s, 40s, 50s, and 60 years or higher) or sex. An age-stratified analysis was conducted because interaction terms indicated statistical significance (the *p*-values of interaction terms were 0.029, 0.065, 0.062, and 0.13 for those in their 30s, 40s, 50s, and 60 years and higher, respectively). In contrast, the *p*-value for the interaction term between sex and COVID-19 infection was 0.808. Logistic regression analysis was performed to examine the associations between COVID-19 infection and HRQoL changes overall and by age category (Model 1: crude model). We adjusted for age and sex in Model 2 and added other potential confounders such as antihypertensive drug use, insulin injection or hypoglycaemic drug use, and cholesterol-lowering drug use in Model 3. In Model 4, we further adjusted for BMI, smoking status, frequency of alcohol consumption, exercise habits, and sufficient rest through sleep in the multiple logistic regression analysis. For the multivariate logistic regression models, odds ratios (ORs) and 95% confidence intervals (CIs) were estimated for ‘worse than baseline HRQoL’ (i.e., negative changes) and ‘not worse than baseline HRQoL’ (i.e., no changes or positive changes) associated with COVID-19 infection. A *p*-value less than 5% (two-tailed) was set as an a priori statistical significance level. All analyses were performed using Stata version 16.1 (Stata Corp LP, College Station, TX, USA).

## 3. Results

Table 1 shows the participants’ characteristics and the proportions of positive, negative, or no changes in HRQoL overall and HRQoL stratified by the presence of COVID-19 infection. Of the 13,365 study participants, 711 (5.3%) developed COVID-19 infection between June 2020 and December 2021. Of those who developed COVID-19 infection, 29.0% reported negative changes in HRQoL, whereas 28.6% and 42.5% reported positive and no changes, respectively. Among those who did not contract COVID-19, 25.2% reported negative changes, and 26.8% and 48.1% showed positive and no changes, respectively. Compared with those who did not develop COVID-19 infection, those who developed COVID-19 infection were more likely to report negative changes in HRQoL in their 30s, 40s, and 50s. Furthermore, those with COVID-19 were more likely to be younger, have a more normal body weight, consume alcohol (daily, rarely), and use antihypertensive drugs, insulin injections, or hypoglycaemic drugs.

Table 2 illustrates HRQoL scores at baseline and 18 months later among 13,365 study participants. Mean (SD) HRQoL was 0.906 (0.101) at baseline and 0.904 (0.109) 18 months later in those with COVID-19 infection. In contrast, those without COVID-19 infection showed higher HRQoL at both baseline (0.918) and 18 months later (0.920) compared with those with COVID-19 infection. A lower mean HRQoL was observed among those in their 30s, 40s, and 50s compared with that observed among those less than 30 years or 60 years or higher of age 18 months later. The changes in HRQoL between baseline and 18 months later in those with COVID-19 infection were −0.0242, −0.0088, and −0.0002 in those in their 30s, 40s, and 50s, respectively. A negative change in HRQoL in participants without COVID-19 infection was only observed in those in their 40s (−0.0026).

Crude and adjusted ORs for negative changes in HRQoL associated with COVID-19 infection are shown in Table 3. We identified statistically insignificant associations between COVID-19 infection and negative changes in HRQoL overall (OR 1.10; 95% CI, 0.91–1.33). However, a statistically significant association was observed in those in their 30s in the fully adjusted Model 4 (adjusted OR 1.70; 95% CI, 1.03–2.81). The corresponding ORs (CIs) for the remaining age groups (i.e., less than 30 years, 40s, 50s, 60 years or higher) were 0.54 (0.15–1.92), 1.14 (0.82–1.57), 1.05 (0.77–1.42), and 0.87 (0.46–1.64), respectively.

Table 4 illustrates five dimensions of HRQoL overall and HRQoL in those in their 30s. Importantly, a higher dimension value signifies lower QOL. Among the five dimensions of HRQoL, mobility, self-care, usual activities, and anxiety/depression worsened; this result was attributed to lower HRQoL in those with COVID-19 infection. In those without COVID-19 infection, declines in QOL (i.e., higher HRQoL scores) were observed in mobility, pain/discomfort, and anxiety/depression. HRQoL scores in every dimension worsened in participants in their 30s with COVID-19 infection, whereas worsening was observed in only self-care, usual activities, and anxiety/depression in those in their 30s without COVID-19 infection.

## 4. Discussion

Our findings suggest that the association between COVID-19 infection and negative changes in HRQoL could be modified by age but not by sex. A 1.7-fold increase in the odds of negative changes in HRQoL was observed only in those in their 30s even after adjusting for possible confounding factors. In contrast, no statistically significant associations were observed in the other generations.

No previous study has reported an association in only those in their 30s, although a Swiss cohort study in a younger population (median age: 43 years) in an outpatient setting indicated the substantial impact of long COVID-19 on HRQoL after mild or moderate acute COVID-19 infection [33]. Our results indicated that HRQoL scores in every dimension worsened with COVID-19 infection in participants in their 30s. This finding is partially consistent with that of a previous study that supported a negative association between younger generations (14–24, 25–38 years of age) and psychological distress after the first pandemic wave [34]. In addition to psychological aspects, problems with usual activities and pain/discomfort were more likely to be caused by long-term COVID-19 symptoms [16]. Similar findings in the distribution of pain/discomfort and anxiety/depression responses were also reported in patients with mild acute infection [35,36,37]. However, these findings were not limited to younger generations. In contrast to the results of this study, a few studies have reported a trend in post-acute COVID-19 syndrome that can significantly worsen QOL among older people [15,38]. These studies mainly focused on patients who were hospitalised during acute COVID-19 infection. Hospitalised patients are older (mean age: approximately 60 years) [15,38], and increased age is associated with weakened immunity. Therefore, age may be an important risk factor for post-acute COVID-19 syndrome. Another cross-sectional study among relatively young health workers (median age: 51 years) identified older age (>49 years) as a risk factor for persistent symptoms that were a long-term consequence of COVID-19 infection [16]. These results might show the possibility of a U-shaped relationship between COVID-19 infection and HRQoL; however, evidence of the association between COVID-19 infection and HRQoL in the younger population remains sparse. The COVID-19 pandemic required people to change their lifestyles and social relationships in potentially different ways by age in addition to the direct health effects due to COVID-19 infection. Therefore, further studies that consider more comprehensive aspects such as socioeconomic status (e.g., educational background, loss or reduction in household income), social support in the community, and workplace social capital are needed to elucidate the relationship between COVID-19 infection and HRQoL.

In contrast to most of the previous findings, no sex differences were observed in the association between COVID-19 infection and HRQoL in this study; interaction terms for COVID-19 infection and sex were not statistically significant (*p* = 0.808). Men represented 74% of our study population, whereas the proportion of women exceeded that of men or women represented most of the study population in the previous studies that showed a higher risk of post-COVID-19 effects among women [16,18]. A large academic COVID-19 hospital in Rome conducted a prospective study of patients who were observed 2 years after hospital admission for severe COVID-19; the study reported that female sex, unemployed status, and chronic comorbidities were the most common predictors of unfavourable values in each EQ-5D-5L domain [9]. However, a single-centre cohort study in Switzerland that included a large proportion of female healthcare workers (75.4%) reported no association between female sex and long symptoms at 90 days after COVID-19 diagnosis [39]. Therefore, the issue of the presence of sex differences remains inconclusive, and further studies are warranted to understand differences in the relevant factors (e.g., marital status, occupation) that affect HRQoL which may produce different volumes of tasks within the family and in the workplace.

A strength of our study was its large number of study participants (13,365 employees and their non-working dependents). A limited number of studies on COVID-19 infection and HRQoL changes from baseline are currently available. However, several study limitations should be noted. First, in contrast to data from medical charts, claims data are collected administratively for reimbursement purposes and are subject to inaccuracies. Specifically, we did not directly evaluate COVID-19 infections; we assumed that a participant was infected when the corresponding ICD-10 codes (U07.1 and U10) appeared in the claims data. Because re-linking claims data for secondary purposes to the original data stored in medical records is not permitted, adjudication of COVID-19 infection via chart review was not feasible. However, exposure misclassification was random (non-differential) for negative changes in HRQoL, and thus the attenuation (towards the null) of the point estimates (i.e., ORs) was possible. Second, the study participants were only those who responded to the additional questionnaires. If healthier people responded to the questionnaires, the results of the study may have been underestimated because of selection bias. Third, the study population consisted of employees and their non-working dependents who responded to the questionnaires via the kencom application. The retirement age (60 years old) in Japan and the fact that some participants withdraw from the health insurance association after retirement may explain why the proportion of those aged 60 years or more was relatively small (13.4%). Similarly, those aged less than 30 years represented only 3.7% of the study population. Younger individuals such as those less than 30 years old tend to be physically healthy and are less likely to use the kencom application to manage their health status. Therefore, the generalisability of the study findings to the overall population is limited.

## 5. Conclusions

This study demonstrates a modified association by age between COVID-19 infection and negative changes in HRQoL: a 1.7-fold increase in the odds of negative changes in HRQoL in those in their 30s. However, in contrast to the findings of previous studies, no sex differences were found. Further studies that consider socioeconomic status such as educational background, loss or reduction of household income, job type, social support, workplace social capital, and work-life balance are needed to deepen the understanding of age and sex differences in the evaluation of the impact of COVID-19 infection on HRQoL among employees.

## Figures and Tables

**Table 1 ijerph-21-00217-t001:** Characteristics of the study participants at baseline in June 2020.

			COVID-19 Infection		COVID-19 Infection	
			Yes		No	
Characteristics	*N*	%	*N*	%	*N*	%
All	13,365	100	711	5.3	12,654	94.7
Age (years; mean, SD)	49.8	9.71	48.2	9.64	49.8	9.71
less than 30	493	3.7	29	4.1	464	3.7
30s	1734	13.0	120	16.9	1614	12.8
40s	3987	29.8	218	30.7	3769	29.8
50s	5360	40.1	267	37.6	5093	40.3
60 or more	1791	13.4	77	10.8	1714	13.6
Sex						
Male	9934	74.3	563	79.2	9371	74.1
Female	3431	25.7	148	20.8	3283	25.9
BMI (kg/m^2^; mean, SD)	22.9	3.4	23.1	3.52	22.9	3.39
less than 18.5	763	5.7	40	5.6	723	5.7
18.5 to 25	8660	64.8	477	67.1	8183	64.7
25 or larger	2791	20.9	151	21.2	2640	20.9
Missing	1151	8.6	43	6.1	1108	8.8
Smoking status						
Yes	1272	9.5	68	9.6	1204	9.5
No	10,580	79.2	570	80.2	10,010	79.1
Missing	1513	11.3	73	10.3	1440	11.4
Frequency of alcohol consumption						
Daily	2684	20.1	160	22.5	2524	20.0
Sometimes	4167	31.2	221	31.1	3946	31.2
Rarely	3679	27.5	204	28.7	3475	27.5
Missing	2835	21.2	126	17.7	2709	21.4
Exercise habits						
Yes	3337	25.0	169	23.8	3168	25.0
No	7189	53.8	416	58.5	6773	53.5
Missing	2839	21.2	126	17.7	2713	21.4
Sufficient rest through sleep						
Yes	6904	51.7	354	49.8	6550	51.8
No	3612	27.0	231	32.5	3381	26.7
Missing	2849	21.3	126	17.7	2723	21.5
Antihypertensive drug use	1680	12.6	110	15.5	1570	12.4
Missing	1515	11.3	73	10.3	1442	11.4
Insulin injection or hypoglycaemic drug use	424	3.2	27	3.8	397	3.1
Missing	1516	11.3	73	10.3	1443	11.4
Cholesterol-lowering drug use	1374	10.3	68	9.6	1306	10.3
Missing	1515	11.3	73	10.3	1442	11.4
HRQoL changes						
Negative changes	3390	25.4	206	29	3184	25.2
No change	6386	47.8	302	42.5	6084	48.1
Positive changes	3589	26.9	203	28.6	3386	26.8
Negative changes in HRQoL by age category						
less than 30 years	113	22.9	3	10.3	110	23.7
30s	448	25.8	41	34.2	407	25.2
40s	1081	27.1	67	30.7	1014	26.9
50s	1389	25.9	79	29.6	1310	25.7
60 or more years	359	20.0	16	20.8	343	20.0

BMI, body mass index; HRQoL, health-related quality of life; SD, standard deviation. Sources of data: health insurance claims, annual health check-ups, and self-reported information from employees and their non-working dependents who used the kencom application (DeSC Healthcare Inc., Tokyo, Japan).

**Table 2 ijerph-21-00217-t002:** Changes in HRQoL scores between June 2020 (baseline) and December 2021 (18 months later) among 13,365 study participants in Japan.

	COVID-19 Infection Yes (*n* = 711)			COVID-19 Infection No (*n* = 12,654)		
	Baseline	18 Months Later	Change in HRQoL Score	Baseline	18 Months Later	Change in HRQoL Score
HRQoL scores, mean (SD)	0.906 (0.101)	0.904 (0.109)	−0.002 (0.105)	0.918 (0.095)	0.920 (0.097)	0.002 (0.095)
Age category						
less than 30 years old	0.900 (0.134)	0.939 (0.083)	0.0397 (0.1080)	0.926 (0.096)	0.928 (0.106)	0.0021 (0.1121)
30s	0.933 (0.090)	0.908 (0.121)	−0.0242 (0.1120)	0.918 (0.100)	0.920 (0.101)	0.0017 (0.1054)
40s	0.900 (0.094)	0.891 (0.117)	−0.0088 (0.1094)	0.918 (0.098)	0.916 (0.100)	−0.0026 (0.0954)
50s	0.902 (0.103)	0.901 (0.103)	−0.0002 (0.0964)	0.915 (0.095)	0.919 (0.096)	0.0037 (0.0926)
60 or more years old	0.897 (0.112)	0.924 (0.085)	0.0270 (0.0960)	0.926 (0.085)	0.933 (0.084)	0.0068 (0.0837)

HRQoL, health-related quality of life; SD, standard deviation. Sources of data: health insurance claims, annual health check-ups, and self-reported information from employees and their non-working dependents who used the kencom application (DeSC Healthcare Inc., Tokyo, Japan).

**Table 3 ijerph-21-00217-t003:** Odds ratios for negative changes compared with positive or no changes in HRQoL score associated with COVID-19 infection in Japan.

	Model 1	Model 2	Model 3	Model 4
	OR (95% CI)	OR (95% CI)	OR (95% CI)	OR (95% CI)
Overall	1.21 (1.03–1.43)	1.22 (1.03–1.44)	1.20 (1.00–1.43)	1.10 (0.91–1.33)
Stratified analysis by age category				
less than 30 years	0.37 (0.11–1.25)	0.40 (0.12–1.35)	0.49 (0.14–1.71)	0.54 (0.15–1.92)
30s	1.54 (1.04–2.28)	1.56 (1.05–2.31)	1.60 (1.00–2.58)	1.70 (1.03–2.81)
40s	1.21 (0.90–1.62)	1.24 (0.92–1.67)	1.23 (0.91–1.68)	1.14 (0.82–1.57)
50s	1.21 (0.93–1.59)	1.22 (0.93–1.59)	1.18 (0.89–1.56)	1.05 (0.77–1.42)
60 or more years	1.05 (0.60–1.84)	1.04 (0.59–1.83)	1.00 (0.56–1.80)	0.87 (0.46–1.64)

CI, confidence interval; HRQoL, health-related quality of life; OR, odds ratio. Model 1: crude odds ratio. Model 2: adjusted for age and sex. Model 3: adjusted for age, sex, antihypertensive drug use, insulin injection or hypoglycaemia drug use, and cholesterol-lowering drug use. Model 4: adjusted for age, sex, body mass index, smoking status, antihypertensive drug use, insulin injection or hypoglycaemia drug use, cholesterol-lowering drug use, frequency of alcohol consumption, exercise habits, and sufficient rest through sleep. Age is not adjusted in the analysis by age category. Sources of data: health insurance claims, annual health check-ups, and self-reported information from employees and their non-working dependents who used the kencom application (DeSC Healthcare Inc., Tokyo, Japan).

**Table 4 ijerph-21-00217-t004:** Changes in overall HRQoL scores and HRQoL scores of those in their 30s between June 2020 and December 2021 in Japan.

	COVID-19 Infection Yes			COVID-19 Infection No		
	At Baseline	At 18 Months Later	Scores Changes	At Baseline	At 18 Months Later	Scores Changes
HRQoL scores overall, mean (SD)	0.906 (0.101)	0.904 (0.109)	−0.002 (0.105)	0.918 (0.095)	0.920 (0.097)	0.002 (0.095)
Dimension						
Mobility	0.007 (0.025)	0.008 (0.027)	0.0007 (0.029)	0.005 (0.021)	0.0046 (0.020)	−0.0004 (0.024)
Self-care	0.001 (0.007)	0.001 (0.010)	0.0006 (0.012)	0.001 (0.007)	0.0012 (0.009)	0.0003 (0.010)
Usual activities	0.004 (0.016)	0.005 (0.019)	0.0009 (0.022)	0.003 (0.013)	0.0029 (0.014)	0.00008 (0.016)
Pain/discomfort	0.021 (0.027)	0.021 (0.027)	−0.0008 (0.029)	0.019 (0.026)	0.0186 (0.025)	−0.0004 (0.027)
Anxiety/depression	0.026 (0.041)	0.027 (0.043)	0.0010 (0.041)	0.023 (0.039)	0.0227 (0.039)	−0.0002 (0.040)
HRQoL scores among those in their 30s, mean (SD)	0.933 (0.090)	0.908 (0.121)	−0.024 (0.112)	0.918 (0.100)	0.920 (0.101)	0.002 (0.105)
Dimension						
Mobility	0.003 (0.014)	0.006 (0.027)	0.0030 (0.030)	0.004 (0.021)	0.0038 (0.019)	−0.0003 (0.027)
Self-care	0.0004 (0.004)	0.001 (0.009)	0.0010 (0.010)	0.001 (0.009)	0.0011 (0.010)	0.0002 (0.011)
Usual activities	0.004 (0.013)	0.007 (0.024)	0.0036 (0.025)	0.003 (0.014)	0.0032 (0.015)	0.0004 (0.019)
Pain/discomfort	0.011 (0.021)	0.015 (0.026)	0.0033 (0.025)	0.014 (0.025)	0.0140 (0.023)	−0.0003 (0.028)
Anxiety/depression	0.023 (0.039)	0.032 (0.049)	0.0092 (0.042)	0.030 (0.043)	0.0299 (0.044)	0.0001 (0.045)

HRQoL, health-related quality of life; SD, standard deviation. Sources of data: health insurance claims, annual health check-ups, and self-reported information from employees and their non-working dependents who used the kencom application (DeSC Healthcare Inc., Tokyo, Japan).

## Data Availability

Restrictions apply to the availability of these data. Data were obtained from DeSC Healthcare Inc. and are available [https://desc-hc.co.jp/ 29 January 2024] with the permission of DeSC Healthcare Inc.

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
