# Peer review of "Impact of COVID-19 Infection on Health-Related Quality of Life in the Japanese Population: A Large Health-Insurance-Based Database Study"

_ijerph, 2024, doi:10.3390/ijerph21020217_

Round 1
Reviewer 1 Report (Previous Reviewer 1)
Comments and Suggestions for Authors
The revisions to the manuscript address most critical issues.
Author Response
Dear Reviewer 1,
We wish to express our appreciation to you for helpful comments on our manuscript. We feel the comments have helped us significantly improve the manuscript. Our responses and descriptions of our revisions are provided as the attached file. We hope that the revised manuscript is now acceptable for publication in International Journal of Environmental Research and Public Health.
Yours sincerely,
Tomoko Kobayashi, on behalf of all authors.

Reviewer 2 Report (Previous Reviewer 3)
Comments and Suggestions for Authors
The authors have produced a clear and well-defined text. They have adequately contextualised the research. However, the authors make the mistake of not going deeper into the conditioning factors of the social context. When they talk about quality of life, it seems that they are referring to a conceptual element that exists independently of reality and that, moreover, is individual. Existing studies clearly show that this is not the case. It is based on individual perception, but that does not mean that it refers to individual elements. I therefore suggest that the authors introduce a small modification in the introduction clarifying these aspects and clearly showing that quality of life is related to the social context.
The methodology is clear and appropriate.
The results are well presented, clear and the tables are easy to understand. However, I suggest that table 1 be modified and made smaller or be divided in two. In addition, all tables must indicate the source of the data.
Author Response
Dear Reviewer 2,
We really appreciate your insightful review and helpful comments for this manuscript (ijerph-2770613). We feel the comments have helped us significantly improve the manuscript. Please find the detailed responses in the attached file and the corresponding revisions/corrections highlighted/in track changes in the re-submitted files.
We hope that the revised manuscript is now acceptable for publication in International Journal of Environmental Research and Public Health.
Yours sincerely,
Tomoko Kobayashi, on behalf of all authors.

This manuscript is a resubmission of an earlier submission. The following is a list of the peer review reports and author responses from that submission.
Round 1
Reviewer 1 Report
Comments and Suggestions for Authors
Authors have not included several similar publications that provide answer to the question raised:
"On multivariable logistic regression analysis (Table 4), we observed that increasing age was significantly associated decline (26–35 years: aOR = 1.5, 95% CI 1.0–2.2; 36–45 years: aOR = 1.9, 95% CI 1.2–2.9; ≥ 46 years: aOR = 2.1, 95% CI 1.4–3.3) in social domain QoL, and females were 1.30 times more likely (aOR = 1.3, 95% CI 1.0–1.6) to have deteriorated social QoL than males during follow-up." [Hawlader, M.D.H., Rashid, M.U., Khan, M.A.S. et al. Quality of life of COVID-19 recovered patients: a 1-year follow-up study from Bangladesh. Infect Dis Poverty 12, 79 (2023). https://doi.org/10.1186/s40249-023-01125-9]
This paper also uses terminology that is not matching with previous presentations; hundreds of papers have been published on this topic; the findings are neither notable nor significant.
Comments on the Quality of English LanguageNot a major issue but substantial changes are recommended.
Reviewer 2 Report
Comments and Suggestions for Authors
Kobayashi et al. presented a paper entitled: "Impact of COVID-19 infection on health-related quality of life in the Japanese population: A large health-insurance-based database study".
The paper is nicely written, nicely organized and clear. However, there are several aspects that could be improved:
Structure: The work is well-organized and rationally constructed, allowing readers to easily follow the information flow. However, consider presenting a quick summary of the important results. Readers can use this as a quick reminder.
Analyses Based on Age: The article clearly illustrates the age-related changes in COVID-19's effects on HRQoL. The emphasis on the greater likelihood of unfavorable changes in HRQoL in people in their 30s is an important discovery. Expand on the possible causes of the observed age-related differences, taking into account aspects such as immunity, post-acute COVID-19 syndrome, and the intensity of symptoms.
Comparison with Previous Studies: The research successfully contextualizes its findings by comparing them to current literature, including studies on post-acute COVID-19 syndrome and its effects on QOL in various age groups. Consider offering a more in-depth comparison with the linked research, noting parallels and contrasts in techniques, demographics investigated, and major outcomes to deepen the conversation.
Sex Differences: The publication adequately acknowledges the lack of sex differences in the current study's link between COVID-19 infection and HRQoL. This discovery is important, particularly in light of past research. Discuss possible explanations for the disparity with previous studies that found female sex to be a risk factor for persistent symptoms, such as differences in research populations, methodology, or healthcare systems.
Limitations: The report recognizes the study's positives, notably the huge number of participants, which lends robustness to the findings. The constraints are thoroughly discussed, however consider expanding on the possible influence of these restrictions on the generalizability of the results.
Conclusion: Based on the shortcomings found in the current study, consider making specific recommendations for future research topics.
Reviewer 3 Report
Comments and Suggestions for Authors
The paper presented is interesting, clear and well constructed. It is a meritorious work that could be published after some modifications.
However, it is important for the authors to understand that much research has been done related to SARS-CoV2 and that it is essential to make the effort to show the innovation and new contributions of the research presented.
Suggestions
I believe that the introduction should be expanded so that the reader of the article can clearly relate the infection to the effects generated by the quality of life. In addition, the reason for choosing the time frame (2020-2021) should be clarified. In this regard, it is essential to explain the social context since, as indicated in the following paper (https://f1000research.com/articles/10-424), there were (at least) two social coping strategies: individualistic and communal. Depending on each of these, quality of life could be affected differently. It is therefore essential that the authors make the effort to explain the social context. Otherwise, quality of life is not adequately understood.
On the other hand, I suggest some small changes that would make the article clearer:
1. after lines 82 and 83 indicate what these codes refer to.
2. In line 101 it is stated that the variables were included as covariates, please justify this decision or at least clarify it further.
3. In the discussion we should explain the reason for doing this analysis after several years have passed and what is new about this work compared to many others that have been done. In line with this, I suggest substantially expanding the search for bibliographical information.
Finally, the conclusions section should be more extensive. In addition, the main highlights of the research should be listed.
I am including a list of papers that are worth reading and even citing by the authors of the research: Daher-Nashif S. In sickness and in health: The politics of public health and their implications during the COVID-19 pandemic. Sociol Compass. (2022) 16:e12949. doi: 10.1111/soc4.12949
Ferreira, L.N., Pereira, L.N., da Fé Brás, M. et al. Quality of life under the COVID-19 quarantine. Qual Life Res 30, 1389–1405 (2021). https://doi.org/10.1007/s11136-020-02724-x
Hansel, T.C., Saltzman, L.Y., Melton, P.A. et al. COVID-19 behavioral health and quality of life. Sci Rep 12, 961 (2022). https://doi.org/10.1038/s41598-022-05042-z
Karakose, T.; Ozdemir, T.Y.; Papadakis, S.; Yirci, R.; Ozkayran, S.E.; Polat, H. Investigating the Relationships between COVID-19 Quality of Life, Loneliness, Happiness, and Internet Addiction among K-12
Teachers and School Administrators—A Structural Equation Modeling Approach. Int. J. Environ. Res. Public Health 2022, 19, 1052. https://doi.org/10.3390/ijerph19031052
Marzo, R.R.; Khanal, P.; Ahmad, A.; Rathore, F.A.; Chauhan, S.; Singh, A.; Shrestha, S.; AlRifai, A.; Lotfizadeh, M.; Younus, D.A.; et al. Quality of Life of the Elderly during the COVID-19 Pandemic in Asian Countries: A Cross-Sectional Study across Six Countries. Life 2022, 12, 365. https://doi.org/10.3390/life12030365
Shek, D.T.L. COVID-19 and Quality of Life: Twelve Reflections. Applied Research Quality Life 16, 1–11 (2021). https://doi.org/10.1007/s11482-020-09898-z
Shek, D.T.L., Leung, J.T.Y. & Tan, L. Social Policies and Theories on Quality of Life under COVID-19: In Search of the Missing Links. Applied Research Quality Life 18, 1149–1165 (2023). https://doi.org/10.1007/s11482-023-10147-2
Vacchiano, M. (2023). How the First COVID-19 Lockdown Worsened Younger Generations’ Mental Health: Insights from Network Theory. Sociological Research Online, 28(3), 884-893.